# Perceptions of receiving behaviour change interventions from GPs during routine consultations: A qualitative study

Chris Keyworth[1]*, Tracy Epton[1], Joanna Goldthorpe[1], Rachel Calam[1], Christopher J. Armitage[1,2]

1 Division of Psychology and Mental Health, Manchester Centre for Health Psychology, School of Health Sciences, Faculty of Biology, Medicine and Health, The University of Manchester, Manchester, United Kingdom, 2 Manchester Academic Health Science Centre, Manchester University NHS Foundation Trust, Manchester, United Kingdom

* chris.keyworth@manchester.ac.uk

**Data Availability Statement:** Illustrative quotations have been provided with the manuscript. Our ethical approval was based upon statements in the participant information sheet and accompanying

## Abstract

General Practitioners (GPs) are expected to provide patients with health behaviour change interventions, yet little is known about the views of patients themselves. We aimed to understand recent patients': (1) general expectations about GPs delivering health behaviour change interventions during routine consultations (including perceptions of appropriateness and helpfulness for receiving interventions), (2) perceptions of responsibility for GPs to talk about health behaviours, and (3) experiences of receiving behaviour change interventions. Twenty-four semi-structured interviews were conducted with people who had recently attended a routine GP consultation. Data were analysed thematically. Three major themes were identified: (1) *acceptability of discussions about behaviour change*, (2) *establishing clinician-patient rapport*, and (3) *healthcare professionals as a credible source and well placed to offer behaviour change* interventions. Most patients were positive about, and were willing to accept behaviour change interventions from their GP during a routine consultation. Although behaviour change was perceived as a sensitive topic for patients, the doctor-patient relationship was perceived to provide an effective platform to discuss behaviour change, with the GP perceived as an appropriate and important healthcare professional from whom to receive advice. Contrary to the views of GPs, behaviour change interventions were perceived by patients as appropriate and helpful during routine medical consultations, particularly where behaviour change could have a positive effect on long-term condition management. Behaviour change interventions delivered by GPs during routine consultations could be used effectively in time-restricted consultations.

## Background

The recent Health Survey for England showed that 87% of adults breached at least some of the national guidance on health behaviour [1]. Smoking, excessive alcohol consumption, sedentariness, unhealthy diet, and overweight/obesity all increase the risk of non-communicable

consent forms that referred only to the use of anonymised quotations. Therefore publication of the full data set is not consistent with our consent statements. These restrictions are in place according to The University of Manchester Research Ethics Committee (ref: 2018-3662-5925). Requests for additional data, that meet the requirements for confidential data access, can be obtained from the Information Governance team at the University of Manchester via information. governance@manchester.ac.uk.

**Funding:** This study was funded by a research grant obtained from Tesco Plc and was supported by the NIHR Manchester Biomedical Research Centre and the NIHR Greater Manchester Patient Safety Translational Research Centre. Tesco had no role in the design of this study and did not have any role during its execution, analyses, interpretation and storage of the data or decision to submit results.

**Competing interests:** The authors have declared that no competing interests exist.

diseases (e.g., cardiovascular diseases, cancers, diabetes) [2], and places a burden on the National Health Service (NHS) [3]. General Practitioners (GPs) are often the first point of contact for patients, and so are well placed to offer brief interventions to support patients to improve their health behaviour. Research has shown that interventions facilitated by a GP in a primary care setting are effective, and are welcomed by patients [4].

Addressing behaviour change with patients is recognised as an important skill to be developed during medical training [5]. Research suggests that the GP-patient interaction provides an important foundation for delivering behaviour change interventions [6–9]. Consistent with these findings is evidence that GPs are delivering behaviour change interventions to at least some patients during routine practice. A recent national survey showed that GPs believed 44% of the patients they saw in a typical week would benefit from a behaviour change intervention, but delivered interventions to just 34% of these patients [10]. Although the data suggest that at least some GPs value this area of clinical practice, there is clearly a mismatch between the numbers of patients needing behaviour change interventions versus receiving such interventions.

One reason for the mismatch between the numbers of patients needing versus receiving behaviour change interventions from their GP concerns barriers to initiating conversations with patients about health behaviour change. These include GP perceptions that patients do not want or need behaviour change interventions [11, 12] and beliefs that patients may lack the motivation and are unwilling to make positive behaviour changes [8, 13, 14]. This results in GPs making conscious decisions about which patients to engage with in discussions about health behaviour [15], based on perceptions of how receptive patients will be to such conversations [16–18].

Research has examined patients' willingness to receive behaviour change interventions within specific health contexts, such as during cancer screening appointments [19] or in the context of managing or preventing long-term health conditions including psoriasis [20] and cardiovascular disease [21], but have not considered consultations for conditions that may have little to do (at least from the perspective of the patient) with the target health behaviour. Consequently, more research is needed on the patient perspective of receiving opportunistic behaviour change interventions from GPs, during routine primary care consultations. Further, it would be valuable to explore whether patient perceptions of receiving behaviour change interventions are consistent with healthcare professionals' views that patients do not want or need information about behaviour change.

There were three specific aims of the present qualitative study, namely, to examine people's: (1) general expectations about GPs delivering health behaviour change interventions during routine primary care consultations (including perceptions of appropriateness and helpfulness for receiving interventions), (2) perceptions of responsibility for GPs to talk about health behaviours, and (3) experiences of receiving behaviour change interventions.

## Methods

### Design and participants

We conducted a qualitative study using semi-structured telephone interviews. Participants, recruited initially through a survey panel company (YouGov), had previously taken part in a large cross-sectional survey examining experiences of receiving behaviour change interventions from a GP during a routine primary care consultation. All participants had consulted with a GP at least once in the four weeks prior to completing the survey.

## Procedure

Ethical approval for the study was obtained from The University of Manchester Research Ethics Committee (ref: 2018-3662-5925). Telephone interviews were conducted between 2018 and 2019. Potential participants, after taking part in the survey, were invited to take part in the qualitative study. We selected a purposive sample with maximum variation to obtain the widest possible views from participants in terms of age, gender and social grade. Those who agreed supplied their contact details so that a member of the research team could arrange informed consent and a convenient time to conduct the interview. Participants provided written informed consent prior to the interviews, which were audio-recorded and transcribed verbatim. Interviews were conducted by one of the study authors (either CK or JG; both trained researchers in conducting qualitative interviews) using a topic guide (presented in Supplementary File A) that probed participants' experiences of receiving behaviour change interventions during routine consultations. It explored three main areas: (1) general expectations about GPs delivering health behaviour change interventions during routine consultations (including perceptions of appropriateness and helpfulness for receiving interventions), (2) perceptions of responsibility for GPs to talk about health behaviours, and (3) experiences of receiving behaviour change interventions. Data collection ceased at the point of saturation, where the research team agreed by consensus that no new themes were emerging from the data.

## Analysis

We took a critical realist stance to analysing the data. This approach assumes that we can capture a reality that exists for the participants in specific contexts, including causal relationships, so long as a reflexive approach is maintained [22]. Due to a lack of existing research, we used inductive thematic analysis to analyse the data. This ensured the findings were data-driven, and we were able to identify and establish patterns across the data[18]. We followed the six phases outlined by Braun and Clarke when conducting thematic analysis [23]: (1) familiarisation with the data through reading the transcripts, (2) generating initial codes relevant to the research question, (3) searching for themes by grouping the codes identified, (4) reviewing themes in relation to coded data, (5) defining and naming themes, and (6) producing the report. One of the study authors carried out the analysis (CK) using NVivo version 12, which was used to organise and manage the data. An appropriate audit trail of the analysis was ensured using field notes [24] after each interview to ensure that interpretations were based on the data collected. We used established quality checking procedures, including critical scrutiny and constant comparison. The lead author shared the coding framework and key illustrative quotes with the other authors as the analysis progressed. Authors discussed any areas of contention and codes and themes were refined accordingly in order to ensure trustworthiness of the data.

## Results

### Sample characteristics

Participant demographics are presented in Table 1 (reported by participants as part of the cross-sectional survey). Social grades were assigned according to the National Readership Survey classification system, whereby the general population are assigned social codes according to occupation: higher managerial and professional (A), intermediate managerial and professional (B), supervisory and professional (C1), skilled manual workers (C2), semi-skilled and unskilled manual workers (D), and casual or lowest grade workers, pensioners, and others who depend on the welfare state for their income (E).

**Table 1. Participant characteristics.**

| Participant ID number | Gender | Age (years) | Ethnicity | Social grade | Smoker | BMI category | Within alcohol guidelines | Meets physical activity guidelines | "5-a-day" | Health behaviour guidelines broached (out of a possible 5) | Long term conditions | Health behaviours discussed at last visit |
|---|---|---|---|---|---|---|---|---|---|---|---|---|
| 41496 | Male | 40 | White British | A | No | Underweight | No | Yes | No | 2/5 | n/a | Alcohol; weight loss. |
| 40265 | Male | 64 | White British | A | No | Obese | No | Yes | Yes | 1/5 | Type 1 Diabetes | Alcohol |
| 40266 | Female | 68 | White British | A | No | Normal weight | Yes | Yes | No | 1/5 | Type 2 Diabetes; Upper gastrointestinal disease; Haemochromatosis | Alcohol |
| 41320 | Male | 67 | White British | B | No | Normal weight | No | Yes | Yes | 1/5 | n/a | Alcohol |
| 40267 | Male | 84 | White British | B | No | Overweight | Yes | Yes | No | 2/5 | Type 2 Diabetes; Arthritis | Alcohol; diet; physical activity; smoking |
| 41300 | Male | 77 | White British | D | No | Obese | Yes | No | No | 3/5 | Underactive thryoid | Alcohol; diet |
| 41916 | Female | 76 | White British | E | No | Overweight | Yes | No | No | 3/5 | n/a | Alcohol |
| 40264 | Male | 35 | White British | A | No | Healthy weight | Yes | No | No | 2/5 | Psoriasis | Alcohol |
| 40268 | Male | 74 | White British | A | No | Overweight | Yes | Yes | Yes | 1/5 | Prostate cancer | Alcohol; diet; physical activity; smoking |
| 40263 | Female | 69 | Any other white background | A | Yes | Normal weight | Yes | No | Yes | 2/5 | COPD | Alcohol; weight loss |
| 41497 | Male | 40 | White British | E | No | Obese | No | No | Yes | 3/5 | n/a | Alcohol; smoking; weight loss |
| 41915 | Female | 44 | White British | C1 | No | Overweight | Yes | No | Yes | 2/5 | Type 2 diabetes; Fibromyalgia | Alcohol |
| 40269 | Male | 71 | White British | C1 | No | Overweight | No | Yes | Yes | 2/5 | Hypertension | Alcohol; smoking |
| 40262 | Female | 73 | White British | B | No | Overweight | Yes | No | Yes | 2/5 | Type 2 Diabetes, Stroke, Degenerative disc disease, Arthritis | Alcohol; smoking |
| 41321 | Male | 61 | Indian | C2 | No | Obese | No | No | Yes | 3/5 | Hypertension | Alcohol; diet; physical activity; smoking; weight loss |
| 40270 | Male | 64 | White British | C1 | No | Obese | Yes | No | Yes | 2/5 | n/a | Alcohol |
| 41914 | Female | 69 | White British | C2 | Yes | Severely obese | Yes | No | Yes | 3/5 | Arthritis | Alcohol |
| 41913 | Female | 55 | White British | C1 | No | Obese | Yes | No | Yes | 2/5 | n/a | Alcohol |

(*Continued*)

**Table 1.** (Continued)

| Participant ID number | Gender | Age (years) | Ethnicity | Social grade | Smoker | BMI category | Within alcohol guidelines | Meets physical activity guidelines | "5-a-day" | Health behaviour guidelines broached (out of a possible 5) | Long term conditions | Health behaviours discussed at last visit |
|---|---|---|---|---|---|---|---|---|---|---|---|---|
| 41301 | Female | 30 | Black African | B | No | Normal weight | Yes | No | Yes | 1/5 | n/a | Alcohol |
| 40271 | Male | 71 | White British | A | No | Overweight | Yes | Yes | Yes | 1/5 | Atrial fibrillation | Alcohol |
| 40272 | Male | 63 | White British | E | Yes | Obese | No | Yes | Yes | 3/5 | Degenerative disc disease; Depression | Alcohol; diet; physical activity; smoking; weight loss |
| 40273 | Female | 70 | White British | B | No | Overweight | Yes | No | Yes | 2/5 | Type 2 diabetes | Alcohol |
| 40274 | Male | 74 | White British | B | No | Overweight | No | Yes | No | 3/5 | n/a | Alcohol |
| 40275 | Female | 70 | White British | E | No | Overweight | Yes | No | Yes | 2/5 | Type 2 diabetes | Alcohol |

The final sample ($n$ = 24), recruited from a total of 493 people who were happy to contacted to take part in the semi-structured interview, included 10 women and 14 men, with variations in age (range = 30–84 years old, mean age = 62) and social grade (A: $n$ = 7; B: $n$ = 6; C1: $n$ = 4; C2: $n$ = 2; D: $n$ = 1; E: $n$ = 4).

Participants' ethnic backgrounds were White British, ($n$ = 21), Indian ($n$ = 1), Black African ($n$ = 1), and other White ($n$ = 1). All participants breached at least one national health behaviour guideline and the sample included three current smokers, seventeen people who said they were overweight or obese, and one person who said they were underweight. Of the total sample, 8 (33.3%) people said they breached national guidelines for alcohol intake, 14 (58.3%) people said they breached national guidelines for physical activity, and 7 (29.2%) people said they did not meet "5-a-day" recommendations for fruit and vegetable intake. Seventeen (70.8%) participants had a long-term health condition. Length of interviews ranged from 15 to 45 min (mean length 25 min).

Three major themes that describe people's perceptions of receiving behaviour change interventions from their GP were identified: (1) *acceptability of discussions about behaviour change*, (2) *establishing clinician-patient rapport*, and (3) *healthcare professionals as a credible source and well placed to offer behaviour change advice*. Illustrative quotes are provided verbatim to illustrate themes, with participant ID number displayed in parentheses.

**Acceptability of discussions about behaviour change.** Participants described the acceptability and expectation of behaviour change discussions with their GP. Participants were particularly receptive to behaviour change interventions if it had a positive impact on an existing health condition, and helped with self-management. Behaviour change was perceived as a sensitive topic, and specific ways of communicating this to patients were suggested.

Most participants wanted to receive information about behaviour change from their GP, in certain circumstances, which was perceived as both helpful and appropriate within the context of a routine GP consultation. Many participants stated that they would welcome a discussion about behaviour change. Participants believed this would give them the autonomy to take control of their own health.

*"From my point of view, very helpful because it takes away the reliance just on medication. It gives you ammunition to try and address issues themselves as opposed to just going to a doctor, getting a tablet and taking the tablet and expecting that to be a universal panacea. From my point of view, it's very helpful."* (40270)

Participants reported that they would expect information about behaviour change from their GP, but particularly in the context of an existing health condition, where behaviour change was seen to improve or to help manage a particular health condition. For some participants however, where behaviour change was outside of the scope of the immediate discussion, behaviour change advice was not always expected.

*"I don't think it happens at every GP appointment that I've had but I think sometimes there is questions about how often you get your. . .because I was diagnosed with hypertension last year and so that became a regular question in terms of, how my exercise is going, how my diet is going and stuff like that."* (41301)

*"I go with a specific problem and that's what I'm there to discuss, you know, yeah, you can have passing remarks about how you are generally, but I wouldn't expect. . .unless it was something that came out from what I was there for. You know, if I was having a lot of say stomach problems, I would expect to be asked what I was eating."* (40263)

Participants described the importance of resources and information that could help with behaviour change. This included resources perceived as helpful for patients and GPs. Examples included the necessary signposting that could be done by GPs to direct patients to the most relevant information about health behaviour change, or better resources in patient waiting areas. Participants described specific ways that information could help them, such as a pre-warning that GPs may initiate a conversation about behaviour change if it was deemed helpful.

*"Yeah, they do run various things and they do have nurse specialists shall we say, so they have the COPD specialist nurse and the lady that does the testing also runs the stop smoking clinic, and then they have a toddler parent group and they have various other, you know, groups that meet on a regular basis for different types of illnesses to discuss and things like that."* (40263)

*"I would suggest that unless there's information sheets put up in a surgery waiting room where a general prompt is delivered where if the doctor thinks you can help yourself, he will advise you, that, as something that they might think about before they get in to see the clinician, I believe may help."* (40268)

Although behaviour change interventions were generally welcomed and expected, participants reported that health behaviour might be perceived by many people as a sensitive topic to discuss. As a result, patients may respond negatively or defensively when GPs raise the topic of behaviour change during consultations.

*"Because it's quite an intimate conversation in a way but obviously when you go to the doctor it's so private but it's quite an intimate thing to say, look, you know, have you considered ways in which you could improve your health yourself."* (40265)

*"I can imagine somebody who perhaps has a different lifestyle might find it inappropriate to be challenged. For example, I can imagine somebody who's perhaps quite a heavy smoker. If the doctor said you need to give up smoking, might find that difficult."* (40270)

**Establishing clinician-patient rapport.** Participants believed that a positive relationship with their GP provided the ideal platform for a discussion about behaviour change. This could be created with continuity of care, regular contact with their GP, as well as GPs communicating relevant and specific information in a two-way collaborative discussion, without the use of "preaching".

The doctor-patient relationship was perceived as an important aspect of being able to have conversations about health behaviour change. Most participants described this relationship, specifically a positive relationship, as providing a platform for facilitating behaviour change conversations. Participants described aspects of trust in the GP as an important component of the relationship with healthcare professionals.

> *"You know, I mean, it's like anything, there are seven GPs at my practice, I think it is, but there are only two that I would, sort of, choose to go and see by choice because they're the two I have the trusting relationship with." (40264)*

Continuity of care was perceived by participants as important to maintaining a positive relationship with their GP. This was particularly true with regular contact over a long period of time. Some participants explained not having regular contact with their GP was problematic and limits opportunities for discussion about behaviour change.

> *"My experience would say that it's not done well by all GPs. It's done by my own GP. And the reason for that is that I've had a long-standing relationship with my GP. And that, I think, is pretty critical. I have friends who tell me that they saw this doctor, that doctor, the other doctor and there was no consistency in the doctors that they saw. And in my case, I've been at the practice longer than the doctor. So I've been with the practice since 1971 and I've seen the practice change hands three times in that time. So the first set of doctors retired, then the next came in and they retired and the next ones came in. But I've had a long-standing relationship and that is, I think, so valuable." (40267)*

Participants believed the way in which information about behaviour change is communicated is important in terms of its acceptability. Building a rapport between the GP and the patient was seen as an important way of ensuring a two-way collaborative discussion about behaviour change. GPs should also try and avoid a "preaching" style of communication, in favour of more patient-tailored approaches.

> *"Yes, I mean, if it's something. . .if the GP appears to be showing genuine sympathy and genuine interest in your problem, then it is very helpful because you would feel that you are receiving support." (40262)*

> *"So instead of being told you must do this, you must do that, if it's like a conversation and it's on, not a paternalistic way, where I'm being told you must to do this but rather like a two-way conversation, I find that more useful." (41301)*

The specific content of health messages communicated to patients was highlighted as an important factor to its acceptability. Participants reported a desire for relevant and specific health information.

> *"I think if. . .regardless of how you're doing it, if it's made relevant to the individual, it's more likely to have an impact. But if it's just general lifestyle, it won't have any impact at all." (40263)*

*"Because it's that personalisation of it, the personal approach, not the sledge hammer of, you know, some cartoon characters telling us to eat five a day but a real person who you respect, who you know and has your health as part of their mission, telling you or discussing with you ways in which you could improve your life."* (40265)

Specific features of communication style were highlighted. Interpersonal skills such as positive reinforcement to raise confidence were highlighted. Ensuring consistency in how health messages are communicated was also emphasised. It was also important to patients that GPs explore specific behaviours that patients want to change.

*"Yes, absolutely. I consider that I. . .I consider that I eat a healthy diet, it doesn't seem to be. . .I find it very difficult to lose weight, even though I'm physically very active. In that case, there's something clearly not right somewhere. It would be helpful to be able to discuss that in some detail."* (40270)

**Healthcare professionals as a credible source and well placed to offer behaviour change interventions.** Participants believed that GPs were an important source of behaviour change interventions, due to them being the first and often only point of contact with the healthcare system. GPs were perceived as trustworthy and being a positive role model, and having the medical training enhanced the credibility of behaviour change messages. Whilst participants described the role of GPs in the prevention of health conditions, and they are well placed to offer advice, participants were also aware of the practical barriers that may prevent this such as high workloads, lack of priority, and time-restricted consultations.

Participants perceived GPs as a credible source of information and valued behaviour change messages that were provided by GPs. Participants perceived GPs as having the expertise to advise patients about the benefits of positive health behaviours on health outcomes. This consequently resulted in messages from GPs, who were perceived as trustworthy and in the opinion of some participants, being perceived as having more weight than similar messages might from other sources. The importance of healthcare professionals, specifically GPs, acting as appropriate role models in terms of their own health behaviours was perceived to enhance the credibility of health messages.

*"Well yes, there must be because it's a voice of authority, isn't it? You'd expect them to know what they're talking about. So perhaps people would take it from the doctor, more than they would from a government minister, for example, or a food company advertising it. So yes, I would think the doctor carried more weight."* (41300)

*"I think modelling that behaviour is very important. If my doctor was a heavy smoker and smelled of whiskey and he was telling me to lead a healthy lifestyle, I would, you know, I'd think, well, you know, take that advice mate."* (40265)

Participants described specific strategies that GPs could focus on when providing behaviour change interventions. First, it was reported that GPs could raise awareness of the importance of positive health behaviours. Second, GPs could offer interventions in terms of improving specific health behaviours. Consequently, this may act as a prompt for behaviour change amongst patients.

*"I think it's. . .in the end, doctors can only give advice, can only offer alternatives and what it does, it's a question of knowledge is power, isn't it. From my point of view, I'd like to engage in the process of making myself better."* (40270)

*"I think they have a duty to themselves and to the NHS to make sure that the patient is aware. I think, ultimately, it is the patient's responsibility, but the patient needs prompting. Now if the patient doesn't take the prompting, then the GP has, I think, a duty to prescribe exercise."* (40275)

Participants also described GPs as being well placed to provide an important role in the prevention of health conditions. One participant spoke about the need to identify important health risk factors that could be addressed to reduce risks associated with other health conditions.

*"But if. . .when he has gone because he had a chest infection, a couple of years ago. Again, it's the GP that picked up his being overweight and said, oh let's see if this is causing you any problems. He might have been caught before he became diabetic, he might have been caught in the borderline stage."* (40266)

*"Yes, because I think it's part of prevention, and the NHS is absolutely brilliant. It saved my life on numerous occasions, but I think, and especially with public health being the way it seems to be, everybody involved in public health needs to be aware of ways of presenting illness and further. You know, we had campaigns on the television and so on about eating healthily and five a day and all the rest of it, it needs reinforcing. . ."*(40265)

However, participants were also aware of the conflicting demands faced by GPs working in primary care. Participants reported that behaviour change may be of low priority to GPs in routine consultations, and they may struggle to incorporate behaviour change discussions in a time-restricted consultation.

*"Well they're very vocal about not going to the doctor and asking about two or three things. And also, I'm aware that they don't really want you to be in there more than 10 or 20 minutes. And of course there is the problem that you're having to wait in order to get an appointment. Where I am, I'm having to wait for a general appointment, not an urgent, about two weeks. So that gives me the impression that the doctor's very busy and I shouldn't waste his time."* (40273)

*"My experience, and this is not meant as a criticism, I understand the pressures on doctors. My experience with doctors generally is that they just don't have time. You've got your 10 minutes, once your ten minutes is gone, that's it."* (40270)

Where participants believed GPs could not always help with health behaviour change due to perceptions of heavy workloads faced by GPs, participants described the importance of referrals, either to other healthcare professionals, or within the local community, as a way of at least accessing *some* behaviour change interventions.

*"Yeah, I think it's also downgrading some. . .that sounds wrong. . .not necessarily the doctor having to do it all. So it could be non-medical people, especially when it comes to lifestyle, it doesn't have to be the doctor, because we're not going to suddenly magic thousands of doctors, but there are a lot of people who are more than capable of filling that role."* (40263)

*"The benefits are, that they could direct people to where they would get specialist advice, because I'm not sure that a GP is a person to be doing that, with all that they are trying to do for their patients. But to direct them to specialist advice. Could act as a bit of a lobbyist in the local area, with what's available to people."* (40266)

*"Yes, I think it is appropriate, but going back to what I mentioned before, I'm not sure it's the best use of highly skilled and highly paid professionals' time that they could, in a sense, my own feeling is they'd come better from a different source, albeit still within the medical, the NHS that could have groups of people. You know, Alcoholics Anonymous, there's bound to be groups like that." (40274)*

## Discussion

This study examined the perceptions of patients receiving health behaviour change interventions from a GP during routine medical consultations. There are three important findings. First, people are generally positive about, and are willing to accept behaviour change interventions from a GP during a routine medical consultation. Second, whilst behaviour change is perceived as a sensitive topic for patients, the clinician-patient relationship is perceived to provide a platform to discuss behaviour change. For most of our participants, the GP is perceived as an appropriate and important healthcare professional from whom to receive interventions. Third, an appropriate communication style, such as making information specific, relevant, and patient-tailored can facilitate the delivery of behaviour change interventions.

### Comparison with existing literature

There are promising findings in the wider literature that suggest patients are willing to accept behaviour change interventions in primary care settings [4, 6], within specific health contexts such as cancer screening [19] or as part of long-term health condition management [20]. Our findings suggest behaviour change interventions may be more widely acceptable to patients during routine clinical interactions with a GP. In the case of weight management, literature focusing on GPs suggests that increased rapport and continuity of care are important factors for increasing the likelihood of behaviour change discussion [8]. These principles are reflected in our findings, and can be applied more generally across all health behaviours. Although the appropriateness of behaviour change interventions during medical consultations is sometimes questioned by GPs [11, 25], our findings suggest patients perceive interventions as both appropriate and helpful within a routine medical interaction. However, this was most prominent in cases where behaviour change was seen to have a positive influence on an existing medical condition, or to the patient's presenting complaint, when a patient had a positive relationship with their GP, and when relevant and specific information was communicated sensitively. Consequently, delivering behaviour change interventions in the context of routine consultations affords the opportunity to address prevention as well as management of health conditions.

Communication style was highlighted as an important factor of delivering behaviour change interventions. Making information relevant and specific to patients was seen to increase the acceptability of health information, which may increase the likelihood of its effectiveness on subsequent behaviour. Observational studies of healthcare professional-patient interactions suggest that there are opportunities during routine consultations to provide more specific health information relating to behaviour change, which is not always acted upon [20, 26]. Additionally, interpersonal skills such as positive reinforcement to raise patients' confidence, and being non-judgemental towards patients, is congruent with the findings of systematic reviews emphasising the benefits of training in these key areas of communication skills [27, 28].

### Implications for practice

Whilst the expectation that GPs should deliver behaviour change interventions is not new, GPs are often ambivalent about their role in providing behaviour change interventions, and

are sceptical of patient receptivity to advice. Our study shows that whilst patients perceive it may not always be possible for GPs to broach behaviour change during consultations, patients would welcome such discussions in particular circumstances, and we report specific ways that advice could be communicated from the perspectives of patients, during routine primary care consultations. A lack of high quality information about behaviour change in primary care settings has been highlighted previously [29], and our study suggests that patients want more accurate, relevant and up-to-date information and resources in the healthcare settings, either in patient waiting areas or from GPs. People in our study suggested this could serve as a prompt either to: (a) discuss behaviour change with their GP, or (b) consider behavioural changes more generally. This is consistent with the broader literature, and specifically with behaviour change techniques such as "prompts/cues", or "restructuring the physical environment" [30] as specific ways of incorporating health information into health care settings. Future research should aim to examine the use of such techniques as a way of facilitating the delivery of health information, particularly given the time restrictions of GP consultations; a recent survey suggests GP consultation rates are an average of 9.22 min [31].

## Strengths and limitations

This study examined people's perceptions of receiving behaviour change interventions during routine GP consultations. Findings strengthen the growing evidence base highlighting important opportunities for GPs to widen the scope of the consultation to deliver interventions as part of routine consultations, with the broad approval of patients, in order to address prevention as well as management of health and long-term condition management. This is particularly true given our entire sample breached at least one national health behaviour guideline; 21 (87.5%) people breached two or more.

There are limitations to the study. Participants in the present study had previously taken part in a cross-sectional survey and were drawn from a pre-existing sample of people who had visited their GP at least once in the preceding four weeks, and had volunteered to be interviewed. Whilst we aimed to capture the widest possible variation of views and opinions, and our sample size was deemed to be sufficient to answer our research questions, there may be additional views that were not captured in the present sample. Further, of the total sample, 17 (70.8%) people had a long-term condition, which may have influenced the responses and receptivity to behaviour change interventions. This is particularly true given the most commonly reported long-term condition (Type 2 diabetes; n = 6 and rheumatoid arthritis; n = 3) are associated with health behaviours. Additionally, the majority of our sample comprised older adults (18 people were over 60 years of age). Although people were receptive to behaviour change interventions, even with the presence of a long-term condition, future research could aim to further examine receptivity to behaviour change interventions in a sample without the presence of a long-term condition.

## Conclusions

Contrary to the views of GPs often reported in the literature, behaviour change interventions is perceived by most patients as appropriate and helpful in the context of a routine consultation, particularly where behaviour change can have a positive effect on long-term condition management. Behaviour change interventions delivered by GPs during routine medical consultations enable interventions to have maximum reach, and can be used effectively when incorporated into time-restricted consultations [4]. A more proactive approach to behaviour change could be adopted in patient consultations with the broad approval of patients.

## Supporting information

**S1 File. Interview topic guide***.
(DOCX)

**S2 File. Consolidated criteria for reporting qualitative studies (COREQ): 32-item checklist.**
(DOCX)

## Author Contributions

**Conceptualization:** Chris Keyworth, Christopher J. Armitage.

**Data curation:** Chris Keyworth.

**Formal analysis:** Chris Keyworth, Tracy Epton, Joanna Goldthorpe, Christopher J. Armitage.

**Methodology:** Chris Keyworth, Tracy Epton, Joanna Goldthorpe, Rachel Calam, Christopher J. Armitage.

**Writing – original draft:** Chris Keyworth, Rachel Calam, Christopher J. Armitage.

**Writing – review & editing:** Chris Keyworth, Tracy Epton, Joanna Goldthorpe, Rachel Calam, Christopher J. Armitage.

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
