## [Editor Report · Decision Letter 0]

24 Jan 2020

PONE-D-20-01897

Perceptions of receiving behaviour change interventions from GPs: A qualitative study

PLOS ONE

Dear Prof Keyworth,

Thank you for submitting your manuscript to PLOS ONE. After careful consideration, we feel that it has merit but does not fully meet PLOS ONE’s publication criteria as it currently stands. Therefore, we invite you to submit a revised version of the manuscript that addresses the points raised during the review process.

See comments below.

We would appreciate receiving your revised manuscript by 24 February. To enhance the reproducibility of your results, we recommend that if applicable you deposit your laboratory protocols in protocols.io, where a protocol can be assigned its own identifier (DOI) such that it can be cited independently in the future. For instructions see: http://journals.plos.org/plosone/s/submission-guidelines#loc-laboratory-protocols

We look forward to receiving your revised manuscript.

Kind regards,

Andrew Soundy

Academic Editor

PLOS ONE

Additional Editor Comments (if provided):

I like the idea. However the methods needs a framework to provide details within sections. Please selected the COREQ (Tong et al) or SRSQ (O'Brien) or another suitable one and make sure the methods addresses the points most important to this type of work. Please make sure you give a methodology and a paradigmatic stance and consider the implications for this on how you have written you results and how you view quality. Please consider aspects such as sampling and sample size, an audit trail within a supplementary file, trustworthiness etc and then resubmit the paper.

Journal Requirements:

2. Please include additional information regarding the interview guide used in the study and ensure that you have provided sufficient details that others could replicate the analyses. For instance, if you developed a guide as part of this study and it is not under a copyright more restrictive than CC-BY, please include a copy, in both the original language and English, as Supporting Information.

4. Thank you for including your ethics statement:  "Ethical approval for the study was obtained from a university ethics committee (ref:2018-3662-5925).".   

---

## [Author Response · Author response to Decision Letter 0]

19 Feb 2020

Please find attached our full response to reviewer comments documents. 

Thank you

---

## [Decision Letter · Decision Letter 1]

14 Apr 2020

PONE-D-20-01897R1

Perceptions of receiving behaviour change interventions from GPs: A qualitative study

PLOS ONE

Dear Dr Keyworth,

Thank you for submitting your manuscript to PLOS ONE. After careful consideration, we feel that it has merit but does not fully meet PLOS ONE’s publication criteria as it currently stands. Therefore, we invite you to submit a revised version of the manuscript that addresses the points raised during the review process.

Please consider the minor changes identified by the reviewer. 

We would appreciate receiving your revised manuscript by May 29 2020 11:59PM. To enhance the reproducibility of your results, we recommend that if applicable you deposit your laboratory protocols in protocols.io, where a protocol can be assigned its own identifier (DOI) such that it can be cited independently in the future. For instructions see: http://journals.plos.org/plosone/s/submission-guidelines#loc-laboratory-protocols

We look forward to receiving your revised manuscript.

Kind regards,

Andrew Soundy

Academic Editor

PLOS ONE

Reviewers' comments:

Reviewer's Responses to Questions

**Comments to the Author**

1. If the authors have adequately addressed your comments raised in a previous round of review and you feel that this manuscript is now acceptable for publication, you may indicate that here to bypass the “Comments to the Author” section, enter your conflict of interest statement in the “Confidential to Editor” section, and submit your "Accept" recommendation.

Reviewer #1: All comments have been addressed

2. Is the manuscript technically sound, and do the data support the conclusions?

Reviewer #1: (No Response)

3. Has the statistical analysis been performed appropriately and rigorously? 

Reviewer #1: (No Response)

4. Have the authors made all data underlying the findings in their manuscript fully available?

Reviewer #1: (No Response)

5. Is the manuscript presented in an intelligible fashion and written in standard English?

Reviewer #1: (No Response)

6. Review Comments to the Author

Reviewer #1: (No Response)

7. PLOS authors have the option to publish the peer review history of their article (what does this mean?). If published, this will include your full peer review and any attached files.

Reviewer #1: No

---

## [Author Response · Author response to Decision Letter 1]

1 May 2020

Please find attached our response to reviewer comments documents. 

Thank you

---

## [Editor Report · Decision Letter 2]

5 May 2020

Perceptions of receiving behaviour change interventions from GPs during routine consultations: A qualitative study

PONE-D-20-01897R2

Dear Dr. Keyworth,

We are pleased to inform you that your manuscript has been judged scientifically suitable for publication and will be formally accepted for publication once it complies with all outstanding technical requirements.

With kind regards,

Andrew Soundy

Academic Editor

PLOS ONE
---

## [Editor Report · Acceptance letter]

12 May 2020

PONE-D-20-01897R2 

Perceptions of receiving behaviour change interventions from GPs during routine consultations: A qualitative study 

Dear Dr. Keyworth:

I am pleased to inform you that your manuscript has been deemed suitable for publication in PLOS ONE. Congratulations! Your manuscript is now with our production department. 

With kind regards,

on behalf of

Dr. Andrew Soundy 

Academic Editor

PLOS ONE